# Cyclodextrin Metal-Organic Framework as a Broad-Spectrum Potential Delivery Vehicle for the Gasotransmitters

**DOI:** 10.3390/molecules28020852

**Published:** 2023-01-14

**Authors:** Li-Guo Liao, Duo Ke, Guo-Chen Li, Sheng Zhang, Bang-Jing Li

**Affiliations:** 1Key Laboratory of Mountain Ecological Restoration and Bioresource Utilization, Chengdu Institute of Biology, Chinese Academy of Sciences, Chengdu 610041, China; 2University of Chinese Academy of Sciences, Beijing 100049, China; 3State Key Laboratory of Polymer Materials Engineering, Polymer Research Institute of Sichuan University, Chengdu 610065, China

**Keywords:** γ-CD-MOF, gasotransmitters, ultrasound-assisted method, delivery carrier, broad-spectrum

## Abstract

The important role of gasotransmitters in physiology and pathophysiology suggest employing gasotransmitters for biomedical treatment. Unfortunately, the difficulty in storage and controlled delivery of these gaseous molecules hindered the development of effective gasotransmitters-based therapies. The design of a safe, facile, and wide-scale method to delivery multiple gasotransmitters is a great challenge. Herein, we use an ultrasonic assisted preparation γ-cyclodextrin metal organic framework (γ-CD-MOF) as a broad-spectrum delivery vehicle for various gasotransmitters, such as SO_2_, NO, and H_2_S. The release rate of gasotransmitters could be tuned by modifying the γ-CD-MOF with different Pluronics. The biological relevance of the exogenous gasotransmitters produced by this method is evidenced by the DNA cleavage ability and the anti-inflammatory effects. Furthermore, the γ-CD-MOF composed of food-grade γ-CD and nontoxic metal salts shows good biocompatibility and particle size (180 nm). Therefore, γ-CD-MOF is expected to be an excellent tool for the study of co-delivery and cooperative therapy of gasotransmitters.

## 1. Introduction

Since Ignarro, Furchgott, and Murad’s Nobel Prize-winning discovery that the endothelial derived relaxation factor (EDRF) was actually nitric oxide (NO), an extensive swell of research into gaseous signaling molecules was inaugurated [1]. Over the last few decades, CO, H_2_S, and SO_2_ have been discovered as new messenger gases [2,3]. These gasotransmitters can be endogenously generated from different metabolisms in mammals and play numerous critical roles in physiological and pathophysiological processes. For example, SO_2_ could be produced in cardiovascular tissues and has physiological effects on the cardiovascular system, including vasorelaxation and cardiac function regulation; NO is related to many physiological pathways, such as platelet aggregation and adhesion, neurotransmission, synaptic plasticity, vascular permeability, hepatic metabolism, senescence, and renal function. What is more, NO also plays a key role in host immunity and tumor suppression [4]. H_2_S plays an important role in the regulation of homeostasis of numerous systems, including the cardiovascular, neuronal, gastrointestinal, respiratory, renal, liver, reproductive, and other aspects [5,6]. Controlled delivery of small doses of modulate gasotransmitters in vivo are promising for numerous biomedical applications. However, the difficulty in storage and controlled delivery of gaseous molecules hindered the development of effective gasotransmitter-based therapies.

In response to the need for controlled gasotransmitter delivery, much work has focused on the synthesis of gasotransmitter prodrugs [7,8,9,10]. Currently, many types of gasotransmitter prodrugs have been designed and studied. Such prodrugs are typically activated by specific triggers, such as light [11,12], pH [13], thiols [14,15], enzymes [16,17,18], or reactive oxygen species [19]. A few gasotransmitter prodrugs showed superior controlled delivery ability. For example, Wang’s group reported clever designs that adopt an intramolecular cycloaddition reaction to release SO_2_ under physiological conditions, and the half-lives of release could be tuned to wide-ranging values by modifying the tether linker [20]. However, there are some obvious limitations in the prodrug strategy. First of all, the synthesis of organic prodrugs is complicated and tedious. Second, one prodrug can only deliver a specific gaseous signaling molecule. Prodrug strategy cannot provide a universal platform for multi-gasotransmitters. Third, many prodrugs activation conditions are relatively harsh, for example, some prodrugs employ UV light or X-ray to trigger gasotransmitter release, which is hard to practically apply due to phototoxicity [21,22]. Furthermore, some prodrugs release gasotransmitters with the production of more or less bioactive parent byproducts that would complicate physiological mechanistic studies of gasotransmitters, for instance, thiol-activated prodrugs. Recently, many polymer-based materials, such as polymer vehicles and dendrimers, are used as delivery vehicles for release of gasotransmitters. However, the unique gas state of gasotransmitters makes them difficult to upload in polymer vehicles. In most cases, they still need to be modified to small prodrugs before encapsulation [23,24]. However, polymer-based gasotransmitters carriers are still short of universality, and their preparations are still complex.

Another important strategy to design gasotransmitters carriers is to capture gas molecules in porous materials including mesoporous silicas [25], zeolites [26], clays [27], and metal-organic framework (MOF) materials [28,29]. The high pore volume, large surface area, and tunable pore size of porous materials allow adsorption of desired gasotransmitters molecules into their pores, and the gasotransmitters can be released locally. Compared with the intricate synthesis process of prodrugs strategy, the adsorption process is relatively facile. MOFs, as a new type of porous coordination compounds, are constructed by the self-assembly of metal nodes and organic linkers and have shown great potential applications in gas storage and separation applications because of their properties of facile and controllable synthesis, structural diversity, porosity, and high specific surface area [30]. In the last decade, a few MOF materials have been explored for the delivery of bioactive gases for medical applications. For example, Pinto et al. prepared a kind of vitamin B3 MOF as potential delivery vehicles for NO [31]; Maldonado and coworkers developed a CO releasing material using MOF [Zn_2_(dhtp)] (dhtp = 2,5 dihydroxyterephthalate) as a host of CO-releasing molecules [32]. However, most MOF materials are composed of heavy metal ions or toxic ligands. The toxicity problem of these MOFs hinders them as gas carriers for therapeutic applications. Furthermore, up to now, all MOF-based gasotransmitter carriers are still short of universality. Developing a safe, facile, time-efficient, and broad-spectrum strategy to deliver multiple gasotransmitters is still a big challenge.

Recently, a kind of porous material built by cyclodextrin (CD) and alkali metal ions, namely γ-CD-MOFs, were reported by Stoddart and co-workers [33]. CDs are a class of cyclic oligosaccharides formed by linking glucopyranose units via α-1,4-glycosidic bonds. The γ-CD-MOFs are body-centered cubic crystals composed of food-grade γ-CDs and alkaline metal ions (K^+^, Na^+^, Rb^+^, Cs^+^). The γ-CD-MOFs have multilevel pores (molecular apertures of 0.78 nm in the structure of γ-CD and larger spherical voids of 1.7 nm in the center of γ-CD-MOFs formed by regular arrangement of γ-CD) and high local concentrations of OH^−^ ions. The intrinsic nature of γ-CD-MOFs provides great potential for drug delivery. Thus, we seek to apply these green materials in storage and delivering of gasotransmitters in this paper.

Herein, we prepared nanometer sized γ-CD-MOF by an ultrasonic-assisted solvothermal method. It was found that γ-CD-MOF could adsorb multiplex gasotransmitters, including SO_2_, H_2_S, and NO, due to its unique properties. The adsorbed gas molecules can be released gradually under physiological condition along with the degradation of γ-CD-MOF. This broad-spectrum gasotransmitter loading capability enables γ-CD-MOFs to serve as carriers for various therapeutic agents. It has been reported that gasotransmitters are intimately connected, and their signaling pathways in vasorelaxation, anti-apoptotic, and anti-inflammatory events are shared [34]. Therefore, the broad-spectrum gasotransmitters payload developed by this study shows great potential as both research tools and therapeutic agents for co-delivery and synergy gasotransmitters.

## 2. Experimental Section

### 2.1. Materials and Characterization

PEG-20000, γ-CD (99%), human hemoglobin, sodium hydrosulfite, and N,N-dimethyl-phenylenediamine sulfate salt were purchased from Sigma-Aldrich (Shanghai, China). Pluronic L31 and Pluronic L61 were purchased from Badische Anilin-und-Soda-Fabrik (Shanghai, China). Plasmid pBR322 DNA, cupric acetate anhydrous (Cu(OAc)_2_), Gelred nucleic acid dye, and DNA loading buffer were purchased from Solaibao (Beijing, China). Potassium hydroxide (KOH), methyl alcohol, and ethyl acetate were purchased from Chengdu Kelong Chemical Engineering Company (Chengdu, China). All other reagents are analytical grade and were used directly without further purification.

Scanning electron microscopy (SEM) was measured by Phenom pro, Phenom-world BV. X-ray powder diffraction (XRD) data were collected on Panalytical B.V at room temperature from 3° to 40° (2θ). Nitrogen gas adsorption–desorption isotherms were performed on a volumetric automatic apparatus (Quantachrome AUTOSORB-1). The UV/Vis results were detected using Varioskan Flash (Thermo, USA). The FT-IR results were determined in Thermo Fisher Scientific Nicolet 6700. The Raman spectra were recorded using a Renishaw Raman Microscope spectrometer with laser emitting at 532 nm. The X-ray photoelectron spectroscopy (XPS) was carried out on Thermo VG Multilab 2000 with Al Kα radiation in an ultra-high vacuum.

### 2.2. Synthesis of γ-CD-MOF

A mother solution was prepared by mixing γ-CD (648 mg) and KOH (224 mg) in pure water (20 mL) with pre-addition of 12 mL MeOH, which was sealed and placed in a glass vessel. The mixed solution was ultrasounded for 5 min, and a clear solution was obtained. Then, 32 mL MeOH and 256 mg of PEG 20000 was added quickly to the reaction solution, and the final solution was then heated at 50 °C for 10 min. Sixty minutes later, the nanosized MOF crystals were collected after separation, washed with EtOH and MeOH twice, and dried overnight at 50 °C under vacuum. In order to remove interstitial solvent, the γ-CD-MOF was immersed in MeOH for one day and the MeOH was refreshed three times, then dried in vacuum at 50 °C for 12 h.

### 2.3. Synthesis of γ-CD-MOF-Pluronics

The γ-CD-MOFs were immersed in a 5% (*v/v*) solution of Pluronic L31 and Pluronic L63 in ethyl acetate, respectively, for 33 h at room temperature. Then, the materials were washed with fresh ethyl acetate followed by filtration (three times) and finally dried overnight at 50 °C in vacuum oven.

### 2.4. SO_2_, H_2_S, and NO Loading into γ-CD-MOF

A defined amount of γ-CD-MOF was weighed and sealed in a flask in an argon-atmosphere glove-box. The sample flask was first evacuated to remove the argon gas and then was back-filled with SO_2_ to a final pressure of 1 atm. The flask with the γ-CD-MOF was kept under SO_2_ atmosphere for 24 h and then the excess SO_2_ was vented off. The absorption processes towards H_2_S and NO of γ-CD-MOF were the same as that of SO_2_.

### 2.5. Detection of SO_2_ Release from γ-CD-MOF and Pluronics Modified γ-CD-MOF

The SO_2_ release was tested by the 5,5′-dithiobis (2-nitrobenzoic acid) (DTNB) test [35]. The solution of DTNB was prepared by first dissolving 0.06 g of DTNB in 10 mL of ethanol and then diluting the ethanol solution to 100 mL with phosphate buffered saline (PBS, pH = 7.4). Disodium ethylene diamine tetraacetate (EDTA) was added to the PBS solution to obtained 1.0 mM EDTA-PBS solution (PH = 7.4). SO_2_-loaded MOF crystals (γ-CD-MOF and γ-CD-MOF-Pluronics) were added to vials containing EDTA-PBS solution. After each pre-designed time interval, 200 μL of supernatant was removed and diluted with EDTA-PBS solution to 4.5 mL. Then 500 μL of DTNB solution was added. After a 15 min reaction, aliquots were transferred to a 96 well plate, and the absorbance values were measured at 412 nm.

### 2.6. Detection of H_2_S Release

Detection of H_2_S release from γ-CD-MOF used methylene blue assay [36]. H_2_S-loaded γ-CD-MOF was added to vials containing phosphate buffer (pH = 7.4). The reaction mixtures were incubated at 37 °C, and 400 μL supernatant was removed at predetermined time points. Then 400 μL of FeCl_3_ (30 mM stock in 1.2 M HCl) and 400 μL of N,N-dimethyl-p-pheneylenediamine sulfate (20 mM stock in 7.2 M HCl) were added to this supernatant 4 μL of Zn(OAc)_2_ (40 mM stock in H_2_O). The reaction mixtures were incubated at room temperature for 30 min to allow the formation of methylene blue complex. When the reaction ended, the aliquots were transferred to a 96-well plate and the absorbance spectra were collected from 550 nm to 800 nm using a microplate reader, and the absorbance values were measured at 676 nm.

### 2.7. Detection of NO

Detection of NO release from γ-CD-MOF used an oxyhemoglobin assay [37]. Lyophilized human hemoglobin was dissolved in 0.1 M phosphate buffer (PH = 7.4), and an excess of sodium dithionite was added to ensure the complete reduction of hemoglobin. Purification and desalting were carried out by passing by the oxyhemoglobin solution over a column of Sephadex G-25. NO-loaded γ-CD-MOF was introduced to a fixed amount of oxyhemoglobin solution. After regular intervals, 200 μL supernatant was transferred to a 96-well plate and quick measured by the microplate reader. The amount of NO molecules released from the γ-CD-MOF was calculated based on the absorbance changes at 401 nm using the literature value of 49 mM^−1^ cm^−1^ for the difference between the molar extinction coefficient of metHb and HbO_2_ at that wavelength (△ε_401 (metHb-oxyHb)_).

### 2.8. DNA Cleavage Assay

Supercoiled DNA cleavage assay was used to evaluate the DNA cleavage potential of various anticancer, antimicrobial, and antiviral compounds and SO_2_ donors [7]. Briefly, pBR322 DNA (1 μg) was incubated in a 13 mg mL^−1^ solution of SO_2_-loaded γ-CD-MOF or γ-CD-MOF-Pluronics or an aqueous solution of NaHSO_3_:Na_2_SO_3_ (3:1 ratio) with the presence of 1 equivalent of Cu(OAc)_2_ in 10 mM Tris-Cl (pH = 8.0). The mixed solution needed reach to a final volume of 20 μL. The solutions were incubated at 37 °C. After a 5 h incubation, the solutions were mixed with DNA loading buffer (4 μL) and immediately subjected to electrophoresis on a 1.0% agarose gel with Gelred nucleic acid dye (4 μL) in TAE (100 mL) using a horizontal slab gel apparatus containing TAE buffer medium under a constant 90 V for 90 min. After electrophoresis, the DNA was visualized by a UV-transilluminator.

### 2.9. Anti-Inflammation Study

RAW 264.7 cells were cultured in complete medium (containing 10% *v*/*v* fetal bovine serum, 1 mM sodium pyruvate, and DMEM, pH = 7.4) at 37 °C in 5% CO_2_ until ~70–80% confluence. The cells were then seeded in the 48-well plate one day before the experiment. Lipopolysaccharide (LPS) was added into the cell culture to initiate the inflammatory response in RAW 264.7 cells and to trigger the expression of cytokines. RAW 264.7 cells were then co-treated with γ-CD-MOF or H_2_S loaded-CD-MOF and LPS (0.5 μg/mL) for 2 h. Thereafter, the cell culture supernatant was collected. The concentrations of TNF-α in the cell culture supernatant were quantified by a commercial ELISA kit.

## 3. Results and Discussion

### 3.1. Synthesis of Nano-γ-CD-MOF for Gasotransmitters

The clinical value of materials on the nanometer scale has become increasingly obvious, particularly in the context of drug delivery [38]. Nanoscale size is essential for drug carriers to avoiding vascular obstruction. Unfortunately, γ-CD-MOF synthesized by the conventional vapor-diffusion method showed a large particle size (micrometer size) [33]. Recently, a modified method with the addition of cetyltrime–thylammonium bromide (CTAB) was reported to produce nanoscale γ-CD-MOF [39]. However, CTAB was quite toxic for cells [40]. In this study, we developed a facile and rapid method for scale-up synthesis of nanoscale γ-CD-MOF, which prepared γ-CD-MOF with the assistance of ultrasonication and introduced PEG 20000 and MeOH as the size modulators. The acoustic cavitation resulting from ultrasonication caused high temperature and increased the cooling rate, thus accelerating the fabrication of γ-CD-MOF. PEG 20000 was added to adjust the nucleation, and MeOH was added to control the crystal growth rate [41]. Compared to original vapor diffusion methods, this method shortens the fabrication process from day-long to minutes. The powder X-ray diffraction (XRD) results revealed high crystallinity of the γ-CD-MOF synthesized by this method, which is consistent with the crystals synthesized by the conventional method in the literature (Appendix A). The SEM image of as-prepared crystals suggested the specific cubic morphology of γ-CD-MOF and its crystal size was about 100 nm (Appendix A). Dynamic light scattering (DLS) results also indicated that the crystal size of as-prepared γ-CD-MOFs was around 180 nm (Figure 1a), which met the requirement of drug delivery very well and enhanced permeability and retention effect (EPR) in vivo.

Biocompatibility is an essential requirement for a material used in drug delivery. The cellular toxicity of γ-CD-MOF was accessed by measuring the cell viability values (Figure 1b) after 24 h using an MTT assay. The results showed that the survival rate remained high (above 95%) even at very high concentration (up to 1.2 mg mL^−1^). In addition, microscope images of cell cultures suggested that, in the presence of these materials, the cell morphology did not show significant differences (Appendix A). This good biocompatibility suggested a good potential application of γ-CD-MOF in the biological field.

The γ-CD-MOF substance displayed a Brunauer–Emmett–Teller (BET) surface area of 778 m^2^ g^−1^ and a total pore volume of 3.284 cm^3^ g^−1^ as determined by N_2_ isotherm at 77 K (Appendix A). Additionally, this material exhibited a very high concentration of functional groups in the pore; there are about one hydroxyl groups in each 100 Å^2^ internal surface area. The described results indicated that γ-CD-MOFs have the potential to be gasotransmitter payloads.

### 3.2. Sulfur Dioxide Adsorption and Release

It has been reported that γ-CD-MOF shows good adsorption capacity towards gaseous molecules, such as carbon dioxide [42] and formaldehyde [43], due to the unique hollow cavities of γ-CD and high local concentrations of OH^−^ ions in γ-CD-MOF structure. Here, we investigated the adsorption and release behavior of three typical gasotransmitters. First, the SO_2_ absorption capacity of as-prepared γ-CD-MOF was studied.

It has been reported that the organic ligand γ-CD has a pore structure with 0.78 nm diameter. This unusual porous structure endowed γ-CD with the ability to trap gas molecules. Song et al. demonstrated that materials with pore size greater than 0.4 nm are essential for the adsorption of SO_2_ by molecule simulation [44]. Therefore, SO_2_ might be adopted in the pore of γ-CD in the γ-CD-MOF framework by host–guest interactions. As expected, the adsorption of SO_2_ on the γ-CD showed an adsorption capacity of 0.37 μmol mg^−1^, indicating that SO_2_ could be loaded in γ-CD pores indeed (Table 1). It is interesting that the γ-CD-MOF showed adsorption capacity of 0.62 μmol mg^−1^, suggesting that the porous structure of γ-CD-MOF also contributed to the SO_2_ adsorption.

Besides the specific porous structure and host–guest interactions between SO_2_ and γ-CD, hydrogen bonding between SO_2_ and hydroxyl groups on the γ-CD-MOF framework might help the SO_2_ adsorption [45]. In order to get insight on the hydrogen bonding between SO_2_ and this material, the IR spectra of the γ-CD-MOF crystals were recorded after adsorption of SO_2_. The IR spectra of as-prepared γ-CD-MOF and SO_2_-loaded γ-CD-MOF are shown in Appendix A. After the material adsorbed SO_2_, the O-H stretching peak read broadened and shifted from 3408 to 3381 cm^−1^, indicating the formation of the hydrogen bond between –OH and O=S=O groups. In order to further study the effect of hydrogen bonding on the SO_2_ loading capability of materials, UiO-66 and UiO-66-(OH)_2_, which have different hydroxyl groups content but similar BET surface area and pore volume, were selected to study their adsorption behavior [46]. The results clearly showed that UiO-66-(OH)_2_ exhibited a higher loading (0.53 μmol mg^−1^) than UiO-66 (0.32 μmol mg^−1^) (Table 1), suggesting that hydroxyl groups were a benefit for the SO_2_ absorption.

In order to study the effect of metal ion on SO_2_ adsorption, we selected γ-CD-MOF-Cs, which has a similar crystal structure to γ-CD-MOF-K, to measure adsorption behavior. As shown in Table 1, γ-CD-MOF-K showed approximately the same adsorption capacity as γ-CD-MOF-Cs, indicating that metal ion did not affect the SO_2_ load capacity of γ-CD-MOF. In addition, the Raman spectra of γ-CD-MOF-K before and after adsorption of SO_2_ further demonstrated that SO_2_ did not interact with the metal ion in the γ-CD-MOF-K framework (Appendix A). In a word, the γ-CD-MOF exhibited good SO_2_ adsorption ability due to the synergistic effect of hydrogen bonding and host–guest interactions.

The γ-CD-MOF showed a poor resistance to water. Their lattices collapse gradually in water [47]. As a result, the gaseous molecules were released during the disruption of γ-CD-MOF. For anti-mycobacterial or anti-atherosclerotic applications, SO_2_ should be released under serum conditions. In this study, the SO_2_ release behavior from the γ-CD-MOF was investigated under physiological conditions (PBS 0.01 M at pH = 7.4, 37 °C) by a DTNB test. The release curves (Figure 2) showed that the vast majority of the SO_2_ was released within the first ten minutes of the experiment. A similar rapidly release behavior was previously observed in MIL-88-B-2OH, which rapidly released all loaded NO in the first 10 min [48]. The SO_2_ was released quickly from γ-CD-MOF in solution since SO_2_ is highly hydrophilic, and the physically adsorbed SO_2_ molecules were instantly released when samples were exposed to solution.

### 3.3. Nitric Oxide Adsorption and Release Studies

The delivery of NO is important for potential impact in the medical and biological areas. Materials based on MOFs and zeolites have been reported as NO vehicles. Unfortunately, the toxicity of most of these materials limited their applicability to a broad range of biomedical uses. The “green” γ-CD-MOF exhibited extremely low cytotoxicity and high specific surface area, which shows greater potential for delivering NO.

It has been reported that hydroxyl groups are beneficial for NO adsorption [49]. Therefore, γ-CD-MOFs may have the ability to absorb NO effectively. As expected, the γ-CD-MOF showed an adsorption capacity of 2.17 μmol g^−1^ for NO. The total amount of NO adsorption is less than that of porous materials and N-diazeniumdiate (NONOate) polymers, but it is comparable to the NO amount adsorbed by HKUST-1 MOF [50].

The release of biologically relevant NO quantity was assessed by an oxyhemoglobin assay. This method is based on the reaction of oxyhemoglobin with NO to form methemoglobin and nitrate [51]. This reaction also illustrated the inhibitory effect of hemoglobin on the biological activities of NO. The NO loaded γ-CD-MOF contacted with oxyhemoglobin solution, and the changes reflected in the UV/vis spectrum of the solution were recorded. As shown in Figure 3a, the changes in the spectrum after 1 h demonstrated the conversion of oxyhemoglobin into methemoglobin. Comparing the initial spectrum with the spectrum obtained after 1 h, decreases in the intensity in the 542 and 577 nm bands were observed, revealing the consumption of oxyhemoglobin. The appearance of the 500 and 630 nm bands indicated the formation of methemoglobin. The shift to lower wavelengths in the band at 415 nm was also consistent with the transformation of oxyhemoglobin to methemoglobin.

The possibility of the oxidation of hemoglobin by other factors was checked by conducting a control experiment with γ-CD-MOF without loading NO. No changes in the spectrum were observed during 1 h (Appendix A), indicating that no oxidation of the oxyhemoglobin occurred when NO was not loaded in the γ-CD-MOF. Thus, the spectral change in Figure 3a was caused by the release of NO from γ-CD-MOF. These results demonstrated that γ-CD-MOFs can deliver NO in a biologically active form.

NO released from γ-CD-MOF in the hemoglobin solution was studied over time. The release behavior was depicted on the NO released amounts versus time. As shown in Figure 3b, γ-CD-MOF reached the maximum release after 1 h, about 2.17 μmol of NO released from 1 g γ-CD-MOF. NO delivered by HKUST-1 (release mass < 1 μmol NO per gram and t_1/2_ < 5 min) has been confirmed to possibly completely inhibit platelet aggregation [49]. Compared with HKUST-1, γ-CD-MOF released more NO, indicating that γ-CD-MOFs can serve as NO delivery vehicles, which have a bright prospect in NO-associated therapies.

### 3.4. H_2_S Adsorption and Release Studies

Exposing γ-CD-MOF to H_2_S resulted in a color change from white to yellow, which revealed the adsorption of H_2_S. The γ-CD-MOF adsorbed almost 0.14 μmol mg^−1^ of H_2_S at room temperature, whereas γ-CD only shows a loading capacity of 0.05 μmol mg^−1^, suggesting that the γ-CD only partly contributes to the H_2_S adsorption of γ-CD-MOF.

It has been reported that the metal ions in MOFs frameworks could serve as adsorption sites for H_2_S, such as CPO-27-Ni [52]. Therefore, we studied the adsorption of H_2_S from the potassium ion in γ-CD-MOF by X-ray photoelectron spectroscopy (XPS). Compared to the pristine γ-CD-MOF, the binding energy of K 2p in H_2_S-loaded γ-CD-MOF exhibited an obvious shift, revealing the bond formation between the K^+^ and H_2_S (Appendix A).

To investigate whether hydrogen bonds were formed between H_2_S and the hydroxyl groups on the γ-CD-MOF framework, we compared the H_2_S adsorption capacities of two MOFs (UiO-66 and UiO-66-(OH)_2_) that have similar structure but different hydroxyl groups contents. As shown in Table 2, UiO-66 showed a higher H_2_S loading than UiO-66-(OH)_2_. The material with less hydroxyl content exhibited higher H_2_S uptake capacity, revealing that the hydroxyl groups on the γ-CD-MOF are not helpful to H_2_S adsorption. In addition, as shown in the IR spectrum for γ-CD-MOF and H_2_S-loaded γ-CD-MOF, the hydroxyl peak in γ-CD-MOF exhibited a red shift from 3408 cm^−1^ to 3423 cm^−1^, which further demonstrates that H_2_S was not incorporated into the γ-CD-MOF by hydrogen bonding (Appendix A).

The release of H_2_S from γ-CD-MOF was carried out by a colorimetric assay in physiological environments where methylene blue formation was monitored. If H_2_S were released, a characteristic UV-visible absorption spectrum with absorption at 676 nm would be expected. When this experiment was performed, as expected, an increasing in absorption at 676 nm was observed, confirming the H_2_S release ability of γ-CD-MOF (Figure 4a). The time course of H_2_S release exhibited a gradually increasing tendency over 1 h (Figure 4b). This release rate is similar to many previously reported H_2_S prodrugs [9,14].

### 3.5. Regulation of Gasotransmitter Payloads and Release Rate

The gasotransmitter prodrugs with different release amounts and release profiles are required to serve different pathologic applications. Accordingly, it was desirable to develop a biocompatible gasotransmitter vehicle to deliver different gasotransmitter amounts at different release rates for a variety of pharmacological applications. The release of gasotransmitter from γ-CD-MOF occurs along with the disruption of γ-CD-MOF in water. Therefore, we envisaged that modifying the hydrophobicity of γ-CD-MOF might regulate the release kinetics of gasotransmitter. It was reported that coating P123 on the surface of MOF materials (MIL-101(Cr) and NIDOBDC) could enhance their water resistance due to the formation of a protective hydrophobic layer [53]. Pluronics are triblock copolymers based on poly (ethylene glycol)-poly (propylene glycol)-poly (ethylene glycol), and their hydrophobicity could be tuned by varying the ratio of poly (ethylene glycol) to poly (propylene glycol). Herein, this kind of copolymers are introduced to the γ-CD-MOFs to tune the hydrophobicity of γ-CD-MOF.

Two Pluronics (L31 (X_PPO_ = 90%, M_w_ = 1100) and L63 (X_PPO_ = 70%, M_w_ = 2650)) were selected to modify γ-CD-MOF. XPS analysis was performed on γ-CD-MOF and Pluronics-modified γ-CD-MOF (γ-CD-MOF-Pluronic) (Appendix A, Appendix A). It was observed that the C/O ratio in the surface of γ-CD-MOF changed after Pluronics modification, indicating that Pluronics coated on the external surface of γ-CD-MOF successfully. The major characteristic peaks of γ-CD-MOF-Pluronics occurred at 5.7°, 7.0°, 13.2°, and 16.9°, which demonstrated that crystallinity is retained after the post-synthetic modification (Appendix A). Appendix A showed the water contact angle of different Pluronics coated γ-CD-MOF. It can be seen that γ-CD-MOF become more hydrophobic after Pluronics modification, and γ-CD-MOF-L63 exhibited more hydrophobicity than γ-CD-MOF-L31.

SO_2_ was chosen as a model molecule to investigate the loading and releasing properties of gasotransmitter-Pluronics modified γ-CD-MOF. As shown in Figure 5, SO_2_ loading capacity and release rates were significantly affected by the molecular weight and hydrophobicity of Pluronics. The SO_2_ adsorption capacity was γ-CD-MOF > γ-CD-MOF-L31 > γ-CD-MOF-L63 (Appendix A). The SO_2_ loading of γ-CD-MOF decreased with the increase in molecular weight of modified Pluronics, suggesting that the SO_2_ loading could be tuned by changing the molecular weight of the Pluronics coating. In addition, the time-dependent SO_2_ release curves for different Pluronics-coated γ-CD-MOF were compared (Figure 5). It can be seen that SO_2_ release rate obviously decreases after Pluronics modification. The release rate was slower when carrying more hydrophobic SO_2_, which confirms that the gasotransmitter release rate was reduced by means of enhancing the hydrophobicity of the γ-CD-MOF to slow the degradation of the material in solution. These results suggest that we can adjust gasotransmitter release rate through tuning the hydrophobicity of the vehicle. The tunability of the gasotransmitter loading and releasing rate expands the application of this method in gasotransmitter-based therapy.

### 3.6. Bioactivity of Gasotransmitter-Loaded γ-CD-MOF

We probed whether the gasotransmitter-γ-CD-MOF could present related physiological effects as demonstrated by other prodrugs. It is known that SO_2_ could induce DNA cleavage and that DNA cleavage ability has been widely used to assess the potential of anticancer and antimicrobial ability of SO_2_ [54]. As shown in Figure 6, γ-CD-MOF-SO_2_, Pluronics-modified γ-CD-MOF-SO_2_ and the positive control sulfite/bisulfate showed two DNA bands, whereas the blank control sample only showed one DNA band. These results indicated that all the loaded SO_2_ carriers show similar physiological effects to sulfite/bisulfate.

The anti-inflammatory effects of H_2_S-γ-CD-MOF were investigated by studying its ability to inhibit LPS-induced TNF-α production in RAW 264.7 cells. The results showed that H_2_S-loaded γ-CD-MOF inhibited TNF-α secretion effectively, and the concentration of TNF-α is similar to the cells without LPS stimulation (Figure 7). Oppositely, the γ-CD-MOFs do not show the same effect, which clearly demonstrated that the inhibition effect on TNF-α production came from the H_2_S released from γ-CD-MOF. Na_2_S exhibited a slight inhibition TNF-α secretion, but its inhibition was far less than H_2_S-loaded γ-CD-MOF.

## 4. Conclusions

We report a facile, safe, and broad-spectrum carrier, Nano-γ-CD-MOF, to delivery various gasotransmitters as needed. As a broad-spectrum carrier, this oligosaccharide-based material could deliver biologically active SO_2_, NO, and H_2_S (0.62 μmol/g SO_2_, 2.17 μmol/g NO, 0.14 μmol/mg H_2_S) effectively, and release the gaseous molecules under physiological conditions. The release rate of can be tuned by coating Pluronics on the surface of γ-CD-MOF. Gasotransmitter-loaded-γ-CD-MOFs exhibited corresponding physiological effects effectively. What is more, the Nano-γ-CD-MOF shows good biocompatibility and particle size (180 nm). Therefore, this broad-spectrum gasotransmitters’ payload is expected to become an excellent tool for the study of co-delivery and cooperative therapy of gasotransmitters.

## Figures and Tables

**Figure 1 molecules-28-00852-f001:**
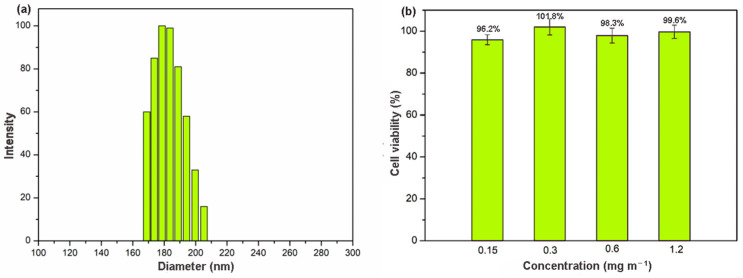
(**a**) DLS result of γ-CD-MOF; (**b**) cytotoxicity of γ-CD-MOF in fibroblast cell line at various concentrations.

**Figure 2 molecules-28-00852-f002:**
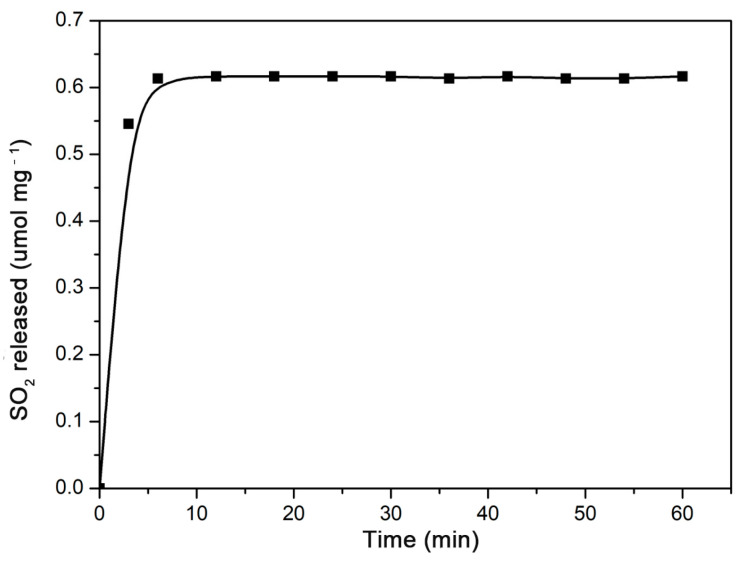
SO_2_ release curve from γ-CD-MOF immersed in PBS (pH = 7.4) at 37 °C.

**Figure 3 molecules-28-00852-f003:**
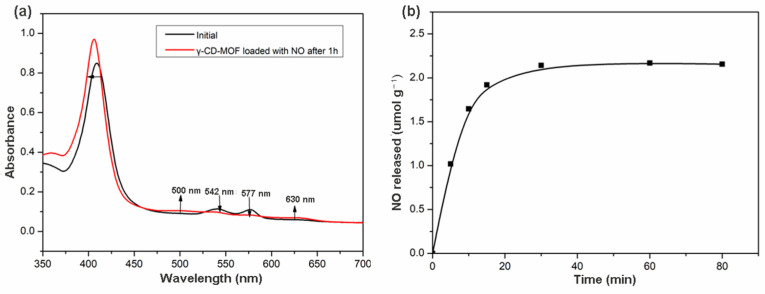
(**a**) Changes in the UV/Vis spectrum of oxyhemoglobin solution upon introduction of NO loaded in γ-CD-MOF; (**b**) NO release profiles in the liquid phase (oxyhemoglobin solution in phosphate buffer) of γ-CD-MOF at 25 °C.

**Figure 4 molecules-28-00852-f004:**
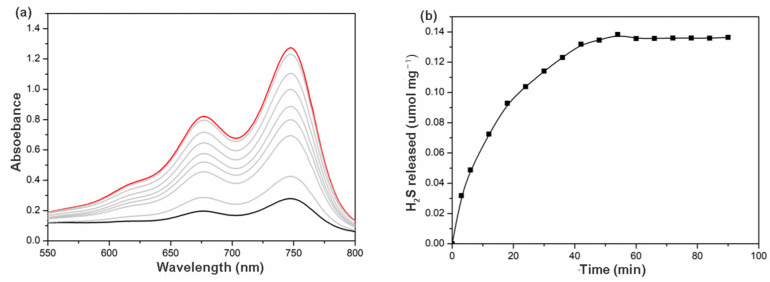
(**a**) Absorbance spectra from the methylene blue assay at different time intervals (the black line/blue lines/red line: 0/0–60 min/60 min); (**b**) H_2_S release curves of γ-CD-MOF in PBS (pH = 7.4) at 37 °C.

**Figure 5 molecules-28-00852-f005:**
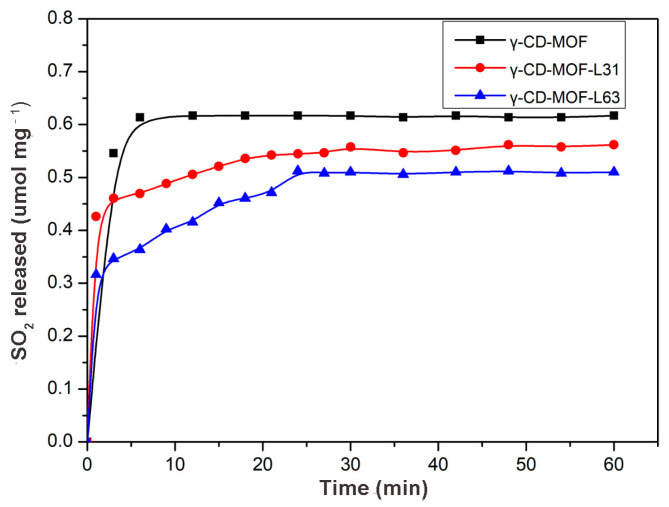
SO_2_ release profiles of different Pluronics-modified γ-CD-MOF in PBS (PH = 7.4) at 37 °C.

**Figure 6 molecules-28-00852-f006:**
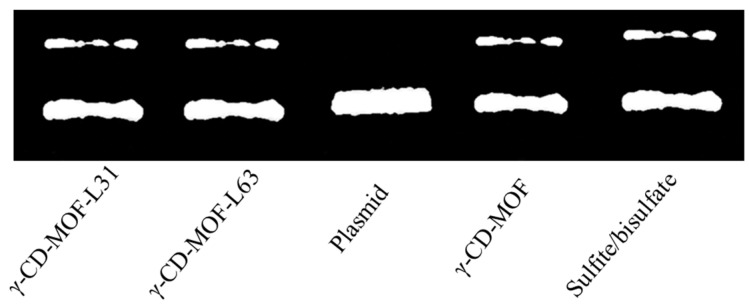
DNA cleavage assay for SO_2_/γ-CD-MOF (13 mg mL^−1^), SO_2_/Pluronics modified γ-CD-MOF (13 mg mL^−1^), and sulfite/bisulfate (1:3, 1 mM).

**Figure 7 molecules-28-00852-f007:**
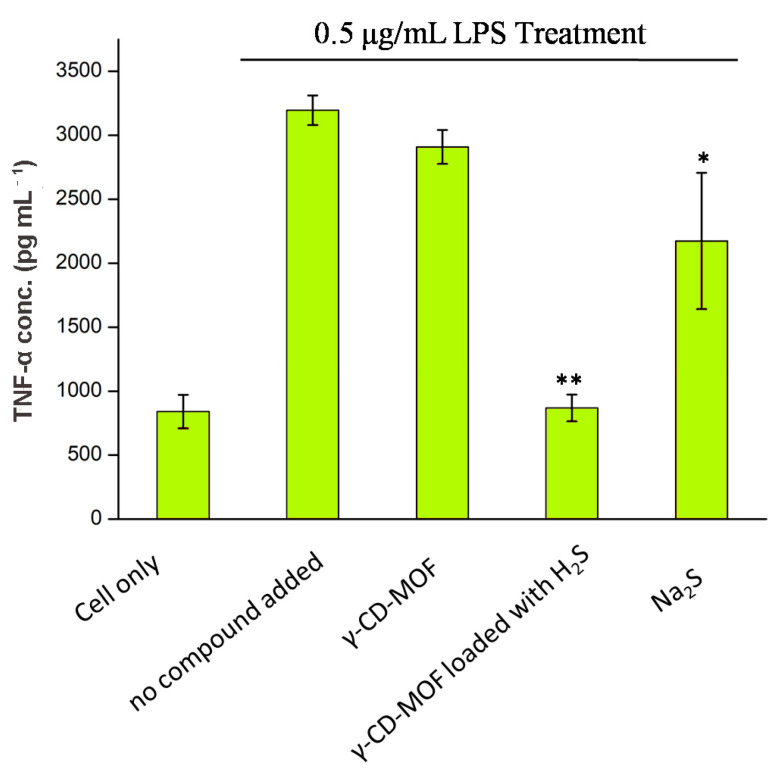
TNF-α concentration of RAW 264.7 cell culture media after 2 h co-treatment with LPS and samples (5 mg mL^−1^ γ-CD-MOF and γ-CD-MOF loaded with H_2_S and 50 μM Na_2_S; “*” and “**”: weak/strong inhibitory activity).

**Table 1 molecules-28-00852-t001:** Adsorption capacity of various materials for SO_2_.

Materials	SO_2_ Adsorption Capacity (μmol/mg)
γ-CD-MOF-K	0.62
γ-CD	0.37
UIO-66-(OH)_2_	0.53
UIO-66	0.32
γ-CD-MOF-Cs	0.64

**Table 2 molecules-28-00852-t002:** Adsorption capacity of various materials for H_2_S.

Materials	H_2_S Adsorption Capacity (μmol/mg)
γ-CD-MOF-K	0.14
γ-CD	0.05
UIO-66-(OH)_2_	0.004
UIO-66	0.009

## Data Availability

The data present in this study are available in insert article or Appendix A here.

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
