# Peer review of "Cyclodextrin Metal-Organic Framework as a Broad-Spectrum Potential Delivery Vehicle for the Gasotransmitters"

_molecules, 2023, doi:10.3390/molecules28020852_

Round 1
Reviewer 1 Report
In this manuscript, Liao et al. describes using nanometer sized γ-CD-MOF as a safe, facile, and wide-scale carrier to delivery multiplex gasotransmitters. By coating different pluronics, the release rate of gasotransmitters can be tuned. The authors give detailed descriptions from background, materials and methods, to results discussion. This paper is complete and has clear logic. The work can be considered for the publication in this journal after revising the following minor issues:
1. Page 2, line 76: should be “Pinto”.
2. Page 3, line 106: the title of section 2 is wrong.
3. Page 5, line 221: please indicate the full name of “DLS”.
4. Page 5, line 230: should be “field”.
5. Page 6: please add the unit in Table 1.
6. Page 9: please add the time in Figure 4a.
7. Page 11: In Conclusion, the authors described “…this oligosaccharide-based material could deliver biologically active SO2, NO and H2S (1.62 umol/g SO2, 2.17 umol/g NO, 0.14 μmol/g H2S) effectively,…” However, for SO2, the adsorption capacity should be “0.62”. Moreover, the unit is different. The authors used “μmol/g” in Conclusion, but in Page 6 line 254 and in Page 8 Table 2, the units are “umol/mg”.
8. Page 11 line 439: please remove “of”.
Author Response
To reviewer #1
1. Reviewer’s comment:
- Page 2, line 76: should be “Pinto”.; 2) Page 3, line 106: the title of section 2 is wrong.; 3) Page 5, line 221: please indicate the full name of “DLS”.; 4) Page 5, line 230: should be “field”.; 5) Page 6: please add the unit in Table 1.; 6) Page 9: please add the time in Figure 4a.
Response: Thanks for the advice of reviewer. We have changed “Pito” to “Pinto” (Page 2, line 81); changed “the title of section 2 (Results and discussion)” to “Experimental Section”(Page 3, line 111); added the full name of --“Dynamic light scattering (DLS)”(Page 5, line 227); changed “filed” to “field” (Page 5, line 238), and added the unit in Table 1 (Page 6, line 265). What’s more, we have added time in Figure 4a (the specific time interval is shown in Figure 4b), please seeing lines 374-375 and Figure 4a.
2. Reviewer’s comment:
1) Page 11: In Conclusion, the authors described “…this oligosaccharide-based material could deliver biologically active SO2, NO and H2S (1.62 umol/g SO2, 2.17 umol/g NO, 0.14 μmol/g H2S) effectively,…” However, for SO2, the adsorption capacity should be “0.62”. Moreover, the unit is different. The authors used “μmol/g” in Conclusion, but in Page 6 line 254 and in Page 8 Table 2, the units are “umol/mg”.; 2) Page 11 line 439: please remove “of”.
Response: Thanks for the advice of reviewer. We have corrected “SO2, NO and H2S (1.62 umol/g SO2, 2.17 umol/g NO, 0.14 μmol/g H2S)” to “SO2, NO and H2S (0.62 umol/g SO2, 2.17 umol/g NO, 0.14 umol/mg H2S” (Page 12, lines 447-448). In addition, we have remove “of” (Page 12, line 453).

Reviewer 2 Report
Reviewer’s Comments:
The manuscript “Cyclodextrin metal-organic frameworks as potential delivery vehicles for the gasotransmitters” is very interesting work. The important role of gasotransmitters in physiology and pathophysiology deserve to employ gasotransmitters for biomedical treatment. Unfortunately, the difficulty in storage and con-trolled delivery of these gaseous molecules hindered the development of effective gasotrans-mitters-based therapies. The design of a safe, facile and wide-scale method to delivery multiple gasotransmitters is a great challenge. Herein, we use γ-cyclodextrin metal organic framework (γ-CD-MOF) as broad-spectrum delivery vehicle for various gasotransmitters, such as SO2, NO and H2S. The release rate of gasotransmitters could be tuned by modifying the γ-CD-MOF with different pluronics. However, the following issues should be carefully treated before publication.
1. In abstract, the author should add more scientific findings.
2. Keywords: the synthesized system is missing in the keywords. So, modify the keywords.
3. In the introduction part, the introduction part is not well organized and cited references should cite recently published articles.
4. Introduction part is not impressive and systematic. In the introduction part, the authors should elaborate the scientific issues in the delivery vehicles research.
5. Synthesis of Nano-γ-CD-MOF for Gasotransmitters…, The author should provide reason about this statement “In this study, we developed a facile and rapid method for scale-up synthesis of nanoscale γ-CD-MOF, which prepared γ-CD-MOF with the assistance of ultrasonication and introduced PEG 20000 and MeOH as the size modulators”.
6. The authors should explain regarding the recent literature why “DLS results also indicated that the crystal size of as-prepared γ-CD-MOFs was around 180 nm (Figure 1a), which met the requirement of drug delivery very well”.
7. Nitric oxide adsorption and release studies. The author should explain the latest literature “The release of biologically relevant NO quantity was assessed by an oxyhemoglobin assay”.
8. The author should provide reason about this statement, “It was found that γ-CD shows a loading capacity of 0.05 μmol mg-1, suggesting that γ-CD only partly contribute to the H2S adsorption of γ-CD-MOF”.
9. Comparison of the present results with other similar findings in the literature should be discussed in more detail. This is necessary in order to place this work together with other work in the field and to give more credibility to the present results.
10. The conclusion part is very week. Improve by adding the results of your studies.
Author Response
To reviewer #2
1. Reviewer’s comment:
- In abstract, the author should add more scientific findings.; 2) Keywords: the synthesized system is missing in the keywords. So, modify the keywords.
Response: Thanks for the advice of reviewer. We have added the “ultrasonic-assisted preparation (Page 1, line 16)” and “particle size (180 nm) (Page 1, line 22)” to the abstract, and added the “ultrasonic-assisted method” to keywords (Page 1, line 25).
2. Reviewer’s comment:
In the introduction part, the introduction part is not well organized and cited references should cite recently published articles.
Response: Thanks for the advice of reviewer. we have added the latest research results, such as references 6, 10, 12, 21, 22, etc, in the introduction part (Page 13, lines 478-479, 486-487, 490-491, 509-510) .
3. Reviewer’s comment:
Introduction part is not impressive and systematic. In the introduction part, the authors should elaborate the scientific issues in the delivery vehicles research.
Response: Thanks for the advice of reviewer. In the revised manuscript, we added some latest research results and rediscussed the scientific issues in the delivery vehicles research (Page 2, lines 65-68).
4. Reviewer’s comment:
- Synthesis of Nano-γ-CD-MOF for Gasotransmitters…, The author should provide reason about this statement “In this study, we developed a facile and rapid method for scale-up synthesis of nanoscale γ-CD-MOF, which prepared γ-CD-MOF with the assistance of ultrasonication and introduced PEG 20000 and MeOH as the size modulators”. 2) The authors should explain regarding the recent literature why “DLS results also indicated that the crystal size of as-prepared γ-CD-MOFs was around 180 nm (Figure 1a), which met the requirement of drug delivery very well”.
Response: Thanks for the advice of reviewer. We synthesized the Nano-scale (180 nm) γ-CD-MOF for Gasotransmitters because this size of particles can avoids vascular obstruction and to enhanced permeability and retention effect (EPR) in vivo. We have added the reason to the revised manuscript (Page 5, line 230).
5. Reviewer’s comment:
Nitric oxide adsorption and release studies. The author should explain the latest literature “The release of biologically relevant NO quantity was assessed by an oxyhemoglobin assay”.
Response: Thanks for the advice of reviewer. We have added the latest literature to replace previous reference (Page 14, lines 570-571).
6. Reviewer’s comment:
The author should provide reason about this statement, “It was found that γ-CD shows a loading capacity of 0.05 μmol mg-1, suggesting that γ-CD only partly contribute to the H2S adsorption of γ-CD-MOF”.
Response: Thanks for the advice of reviewer. We derived that “γ-CD only partly contribute to the H2S adsorption of γ-CD-MOF” since that γ-CD-MOF adsorbed almost 0.14 μmol mg−1 of H2S at room temperature, but γ-CD shows a loading capacity of 0.05 μmol mg−1. In the later investigation, it has been proved that the metal ions in MOFs frameworks also served as adsorption sites for H2S. In order to make this discussion more clear in the revised manuscript, We have changed the discussion to “γ-CD-MOF adsorbed almost 0.14 μmol mg−1 of H2S at room temperature, while γ-CD only shows a loading capacity of 0.05 μmol mg−1, suggesting that the γ-CD only partly contribute to the H2S adsorption of γ-CD-MOF.” (Page 8, lines 348-349).
7. Reviewer’s comment:
Comparison of the present results with other similar findings in the literature should be discussed in more detail. This is necessary in order to place this work together with other work in the field and to give more credibility to the present results.
Response: Thanks for the advice of reviewer. At present, the literature on gasotransmitters carriers mainly focuses on single gas delivery and there is no report on carriers with multiple gases. In the revised manuscript, we added latest literature as references 6, 10, 12, 21, and 22.
8. Reviewer’s comment:
The conclusion part is very week. Improve by adding the results of your studies.
Response: Thanks for the advice of reviewer. We have added some results in the conclusion section to enrich it (Page 12, lines 451-453).
Reviewer 3 Report
In this manuscript, the authors have successfully reported a facile, safe and broad-spectrum carrier, γ-CD-MOF, to delivery various gasotransmitters as needed. What’s more, the release rate of could be tuned by coating pluronics on the surface of γ-CD-MOF. So, this material can become an effective tool for studying the physiological application of gasotransmitters.
However, there are still some problems (listed below) in this manuscript.
1. Up to now, most MOF-based gasotransmitters carriers are short of universality, while the γ-CD-MOF shows broad-spectrum ability to deliver multiple gasotransmitters. But there exist four kinds of gasotransmitters: SO2, NO, CO and H2S. Has the author done any research on CO?
2. In Table 1, the adsorption capacity of γ-CD-MOF-Cs for sulfur dioxide is 0.64, which is greater than that of γ-CD-MOF-K. Is there progress in studying the influence of metal ions on SO2 adsorption?
3. For references, the author should cite some research progress of gasotransmitters in the last two years.
4. There are some format errors in manuscript, such as “t1/2 < 5 min (line 336)”, “H2S (line 454)” and so on. Please double check it.
In short, I suggest the author can make detailed revisions before publishing.
Author Response
To reviewer #3
1. Reviewer’s comment:Up to now, most MOF-based gasotransmitters carriers are short of universality, while the γ-CD-MOF shows broad-spectrum ability to deliver multiple gasotransmitters. But there exist four kinds of gasotransmitters: SO2, NO, CO and H2S. Has the author done any research on CO?
Response: Thanks for the advice of reviewer. We ever tried to study the adsorption and release of CO on the γ-CD-MOF, and the results showed that poor CO adsorption, which may be due to the weak polarity of CO molecule.
2. Reviewer’s comment:In Table 1, the adsorption capacity of γ-CD-MOF-Cs for sulfur dioxide is 0.64, which is greater than that of γ-CD-MOF-K. Is there progress in studying the influence of metal ions on SO2 adsorption?
Response: Thanks for the advice of reviewer. In the manuscript, we compared adsorption capacity of γ-CD-MOF-Cs with that of γ-CD-MOF-K (0.64 vs 0.62 μmol/mg). It can be seen that different metal ions have little impact on sulfur dioxide adsorption.
3. Reviewer’s comment:For references, the author should cite some research progress of gasotransmitters in the last two years.
Response: Thanks for the advice of reviewer. In the references, we have added some recent research results. For example, References 6, 10, 12, etc.
4. Reviewer’s comment:There are some format errors in manuscript, such as “t1/2 < 5 min (line 336)”, “H2S (line 454)” and so on. Please double check it.
Response: Thanks for the advice of reviewer. We have changed “t1/2 < 5 min”, “H2S ” to “t1/2 < 5 min (Page 8, line 337)” , “H2S (Page 12, line 448)” and so on.